# Therapeutic Alternatives in Diabetic Foot Patients without an Option for Revascularization: A Narrative Review

**DOI:** 10.3390/jcm11082155

**Published:** 2022-04-12

**Authors:** Gerhard Ruemenapf, Stephan Morbach, Martin Sigl

**Affiliations:** 1Vascular Center Oberrhein Speyer-Mannheim, Department of Vascular Surgery, Diakonissen-Stiftungs-Krankenhaus, 67346 Speyer, Germany; 2Department of Diabetology und Angiology, Marienkrankenhaus, 59494 Soest, Germany; s.morbach@hospitalverbund.de; 3Department of Cardiology, Angiology, Haemostaseology and Medical Intensive Care, University Medical Centre Mannheim, Medical Faculty Mannheim, Heidelberg University, European Center for AngioScience (ECAS) and German Center for Cardiovascular Research (DZHK) Partner Site, 68199 Mannheim, Germany; martin.sigl@umm.de

**Keywords:** diabetic foot syndrome, revascularization, conservative treatment, no-option, non-constructible limb, critical limb threatening ischemia, CLTI, major amputation

## Abstract

Background: The healing of foot wounds in patients with diabetes mellitus is frequently complicated by critical limb threatening ischemia (neuro-ischemic diabetic foot syndrome, DFS). In this situation, imminent arterial revascularization is imperative in order to avoid amputation. However, in many patients this is no longer possible (“too late”, “too sick”, “no technical option”). Besides conservative treatment or major amputation, many alternative methods supposed to decrease pain, promote wound healing, and avoid amputations are employed. We performed a narrative review in order to stress their efficiency and evidence. Methods: The literature research for the 2014 revision of the German evidenced-based S3-PAD-guidelines was extended to 2020. Results: If revascularization is impossible, there is not enough evidence for gene- and stem-cell therapy, hyperbaric oxygen, sympathectomy, spinal cord stimulation, prostanoids etc. to be able to recommend them. Risk factor management is recommended for all CLTI patients. With appropriate wound care and strict offloading, conservative treatment may be an effective alternative. Timely amputation can accelerate mobilization and improve the quality of life. Conclusions: Alternative treatments said to decrease the amputation rate by improving arterial perfusion and wound healing in case revascularization is impossible and lack both efficiency and evidence. Conservative therapy can yield acceptable results, but early amputation may be a beneficial alternative. Patients unfit for revascularization or major amputation should receive palliative wound care and pain therapy. New treatment strategies for no-option CLTI are urgently needed.

## 1. Introduction

The incidence of both diabetes mellitus as well as peripheral arterial disease of the lower limb arteries (PAD) is rising worldwide [1,2]. PAD often ends in critical limb-threatening ischemia (CLTI), with pain at rest or tissue loss (Table 1). In Germany, for example, more than 13,000 major amputations are performed every year [3,4] for CLTI, mostly in persons with neuroischemic diabetic foot syndrome (DFS) [4,5].

Arterial revascularization is recommended in CLTI; there is adequate evidence [6,7,8,9] on its ability to relieve pain, heal wounds, and avoid amputations, being successful in approximately 90% of cases [10]. However, revascularization may be technically unfeasible (“no-option”), or the patients may come too late for their limb to be salvaged, or they are too sick to receive any form of operation. Conservative treatment of CLTI (pain control, wound care etc.) has hitherto been regarded the worst option, with a 25% mortality and 50% limb loss after 1 year [7,11].

In this situation, a host of alternative methods for the improvement of arterial perfusion, wound healing and pain therapy are employed, the proof of efficacy as well as the scientific evidence of which are lacking. The objective of this review is to investigate whether there are effective and guideline-compliant methods of treatment for diabetic patients with no-option CLTI, and whether they are more useful than conservative treatment or timely major amputation.

## 2. Methods

The German Society for Angiology (DGA) performed an extensive literature search for the 2015 edition of its evidence-based PAD guidelines [6]. For the present article, that particular literature search has been extended to 2020 using MEDLINE, PubMed, and the Cochrane Database of Systemic Reviews. Search items included: CLTI, chronic limb threatening ischemia, CLTI, diabetic foot syndrome, peripheral arterial disease, PAD, angioplasty, endovascular revascularization, surgical revascularization, stent, pharmacotherapy, no-option CLTI, non-reconstructable limb, major amputation, conservative treatment. Individual publications were checked for previously unidentified studies. Particular importance was assigned to randomized controlled trials (RCTs) and their meta-analyses.

## 3. Epidemiology and Concomitant Diseases

PAD signalizes cardiac and cerebral complications as well as a reduced life expectancy. Risk factors include age, sex, cigarette smoking, arterial hypertension, hyperlipidemia, diabetes mellitus, obesity, lack of exercise, and previous events in other vascular territories. In 2015, more than 237 million people worldwide suffered from PAD [2]. The prevalence stands at 20% of all primary care patients aged over 65 years in Germany [12]. Only 1–3% of these have CLTI [7]. In Germany, 480,000 PAD patients per year are treated in hospitals, 44% of whom have CLTI [13], and most of whom suffer from DFS. Amputation-free survival of no-option patients is 28% after 1 year [11], and the quality of life (QoL) is poor [14]. No-option CLTI patients show a more severe pattern of PAD with distal arterial lesions and worse outcome than CLTI patients who can be successfully treated by revascularization [15].

## 4. Treatment of CLTI

The therapeutic goals for CLTI are pain relief, avoidance of infections, major amputations and care cases, healing of wounds, maintenance of walking function, self-determination and social integration, as well as an increase in the quality and duration of life.

Reducing the progression of PAD and cardiovascular morbidity by lifestyle improvement, smoking cessation, or the administration of ASS and statins come too late for DFS patients with CLTI, while optimal control of blood pressure and diabetes are important in view of imminent operative procedures and wound healing.

Arterial revascularization should be performed soon (“time is leg”; [6,7,8,9]). This is supported by appropriate wound therapy, antibiotics according to smear results, wound debridement, necrosectomy, minor amputations, as well as pain therapy.

Revascularization may be impossible if the patient does not offer any technical option for revascularization (“no-option”), comes “too late” to avoid major amputation, or is “too sick” to indulge it.

“No option”: 20 years ago, 40% of CLTI patients [16] were considered “no-option“(Figure 1 and Figure 2). Today, major amputation can be prevented in 90% of cases, owing to aggressive revascularization [10]. However, only half of the CLTI patients are offered a revascularization, and only 25% had angiography before major amputation [17,18]. Short lesions of the crural arteries are successfully treated by angioplasty, while long occlusions are the domaine of crural or pedal vein bypasses [19]. The secondary 5-year patency rates of the latter are >60%, and the amputation rate during this time span can be reduced by 80% [6,7,19]. These techniques should be considered complementary. Before declaring a patient “no-option”, infrapopliteal arteries should be exposed by a skilled vascular surgeon in order to assess their suitability as receiver segments for a bypass. In diabetic patients with CLTI, a new pattern of arterial occlusions seems to become more frequent [20], namely with involvement of below-the-ankle arteries (40–50%). These patients have a higher risk of revascularization failure than patients with below-the-knee PAD whereby the foot arteries are preserved [20].

“Too late”: Tissue loss is too extensive (Figure 1), bacterial infections are not accessible to antibiotic treatment, dementia-related hypo- or hyper-motility or contractures render wound care impossible.

“Too sick”: The patient is not “fit for operation”, owing to serious comorbidities (for example cardiac, pulmonary).

## 5. Therapeutic Alternatives in No-Option CLTI

Many methods are applied in CLTI patients when arterial reconstruction is no longer possible. This “real world non-evidence practice” [21] arises from the urge to help the patients in cases where traditional methods have failed or are not applicable.

### 5.1. Vasoactive Drugs

There is no proven benefit of vasoactive drugs such as naftidrofuryl, cilostazol or pentoxyphyllin on wound healing and amputation rate in CLTI [6,7,8,9,21]. Prostanoids were believed to relieve pain and improve wound healing as well as amputation-free survival in CLTI patients. However, RCTs did not confirm any positive effect of prostanoids on the major amputation rate as well as on wound healing in CLTI patients [22,23,24]. Actual guidelines do not recommend prostanoids for the treatment of CLTI [21].

### 5.2. Fibrinolytic and Defibrinogenating Agents

Fibrinolytic drugs are supposed to decrease plasma-fibrinogen, plasma viscosity, the aggregation of erythrocytes, and to improve collateralization. However, they do not reduce amputation rates or promote wound healing [25]. In an uncontrolled cohort study on 77 no-option DFS patients receiving intravenous urokinase for 3 weeks, ulcers healed in 33% of the surviving patients within one year. Amputation-free survival was 69% [26]. The evidence of these results is minimal. RCTs have never been performed. Some historical RCTs with the defibrinogenating agent Ancrod in patients with CLTI [27] could not demonstrate any improvement in clinical outcome. Urokinase as well as Ancrod are not recommended for no-option CLTI [21].

### 5.3. Hyperbaric Oxygen Therapy (HBOT)

The effects of HBOT on wound healing are most pronounced when blood flow to the foot is maintained. In CLTI, however, they are virtually undetectable [28]. Earlier meta-analyses indicated positive effects of HBOT on wound healing but not on the amputation rate [29,30,31] in patients with neuroischemic DFS. Current RCTs, however, could not demonstrate any positive effects of HBOT on wound healing and amputation rates in ischemic [32,33] or neuropathic DFS patients [34]. Actual guidelines do not recommend HBOT as an alternative to revascularization in DFS patients with CLTI [21].

### 5.4. Lumbar Sympathectomy (LSE)

LSE (surgical or chemical) may reduce rest pain in CLTI but has no effect on ankle brachial index (ABI), mortality, or the amputation rate [35]. In neuropathic DFS, LSE is not sensical, owing to absence of ischemic pain and since additional vasodilatation is not possible due to autonomic neuropathy. LSE is not recommended for the prevention of amputation [6] and is accepted only for the treatment of CLTI in clinical studies in order to obtain evidence [9]. There are no RCTs for “no-option” CLTI to compare lumbar, laparoscopic or percutaneous sympathectomy with a control group [36]. In conclusion, there is no evidence for LSE in “no-option” CLTI patients [21].

### 5.5. Epidural Spinal Cord Stimulation (SCS)

Evidence for SCS in CLTI in terms of avoiding major amputations is low. SCS does not decrease the amputation rate or mortality of DFS patients with CLTI [37]. Others conclude that SCS can reduce amputation rate and pain in no-option CLTI within 1 year, but cannot improve wound healing [38]. Costs and complication rates are high [37,38]. Only one neurosurgical guideline gives a low-level recommendation [39], while vascular guidelines [7,21,35] do not.

### 5.6. Gene and Stem Cell Therapy

Much hope rests on regenerative (gene and stem cell) formation of new capillaries from progenitor cells, angioblasts or pre-existing vessels.

Although the results of gene-based therapy were initially encouraging, they have so far been contradictory and questionable [40]. A placebo controlled RCT (TAMARIS) with 525 patients (53% diabetic) showed that the non-viral 1 fibroblast growth factor NV1FGF neither reduces the amputation rate nor the mortality of CLTI patients [40,41]. Therefore, evidence for the effectiveness of gene therapy in no-option CLTI is lacking.

Mononuclear bone marrow stem cells may promote the formation of collaterals in patients with no-option CLTI. Previous studies with intramuscularly administered bone marrow stem cells showed a significant decrease in resting pain and amputation rate as well as an improvement in the ABI and foot ulcer healing rate. None of these positive results in no-option CLTI remains significant if the analysis is limited to randomized, placebo-controlled studies or low-bias RCTs [42]. The placebo-controlled JUVENTAS RCT study demonstrated the ineffectiveness of repetitive intraarterial administration of mononuclear bone marrow cells on the amputation rate [43]. Some secondary outcomes were also improved under the placebo. Both the efficacy as well as the evidence of the costly cell therapy in no-option CLTI is low [44].

### 5.7. Hypertensive Extracorporeal Limb Perfusion (HELP)

HELP aims at the formation and expansion of collaterals to prevent major amputation in no-option CLTI patients through shear stress and increased wall tension. The procedure comprises isolated arterial perfusion of the affected limb with a heart-lung machine at 200–300% of the normal blood pressure [45]. Amputation was prevented in selected cases. Nothing has been heard about the method since 2013. Evidence has never reached said goal.

### 5.8. Intermittent Pneumatic Compression (IPC) 

IPC is a procedure that was first described some 90 years ago (PAVEX; [46]). It was intended to reduce the amputation rate, ease pain and accelerate wound healing in ‘no-option’ CLTI. Several retrospective case series as well as a few controlled studies (for example, [47]; 47% diabetes patients) were performed in CLTI patients using different devices and study protocols, with positive effects on wound healing and amputation rate. However, RCTs have never been performed; therefore, there is no substantial evidence [48] to recommend IPC for no-option diabetic patients. IPC is not mentioned in the current guidelines.

### 5.9. Transcutaneous Muscular Electrostimulation (TES)

TES may promote angio- and vasculo-genesis. In an RCT with 22 patients with no-option CLTI [49], an improvement in flow velocity in the anterior tibial artery was found when TES was combined with prostacyclin infusions. Prostacyclin alone had no such effect. The oxygen saturation in the tissue did not change in any group. Since the pain-free walking distance of all patients increased from 89 to 300 m after 2 weeks of treatment, none can have suffered from no-option CLTI. There is no evidence for TES in no-option CLTI. Moreover, there is no evidence for extracorporeal shock wave treatment (ESWT) in no-option CTLI.

### 5.10. Special Wound Dressings

After extensive minor amputations of no-option diabetic patients, major amputation could be avoided by skin replacement material stimulating the synthesis of extracellular matrix from fibroblasts and creating a ‘neodermis’ [50]. The study included 26 patients. The results are not scientifically substantiated.

### 5.11. Ozone Therapy

Immunomodulation with ozonated autologous blood or oils to increase vasculo-genesis in patients with no-option CLTI was repeatedly shown in small observational studies. Ozone therapy is considered irrational and life-threatening because such high doses of ozone significantly increase the risk of neoplasia [51]. There is no evidence of ozone treatment in no-option CLTI.

### 5.12. Arterialisation of the Deep Leg Veins (DVA or Deep Vein Arterialisation)

Attempts have been made to arterialize the deep veins of the lower leg and foot [52] in no-option patients using a bypass. This is usually followed by rapid occlusion, owing to venous valves and high intravascular resistance impeding arterial flow.

The model has been rediscovered as “percutaneous arterialization of the deep leg veins” or a combination of open bypass surgery on deep leg and foot veins with intraoperative destruction of the venous valves. In particular, open surgery showed leg preservation and wound healing between 25 and 100%, primary patency between 44 and 88%, and the elimination of resting pain between 12 and 100% [53]. Midterm outcomes indicate that the concept might prevent major amputation in some CLTI patients [54]. To date, no guideline mentions this treatment for no-option CLTI.

### 5.13. Conservative Treatment

The aims are to avoid pain, wound infections, hospitalization and amputation in patients unfit for revascularization or amputation, or for no-option patients.

“Real-world” data show that CLTI patients without revascularization have an extremely low amputation-free survival [18]. Recently, however, more favorable results were reported [7,11,55], resulting in amputation rates of 13 [19] to 19% after 6 months and 23% after 1 year if the wound treatment is optimal and soft tissue infections can be controlled [55]. In 602 DFS patients with CLTI who did not receive revascularization, 50% of the wounds healed, 17% of the patients needed a major amputation, and 33% died within 1 year [56]. Conservative treatment was assumed by some not to be significantly worse than revascularization in terms of survival and amputation-free survival [49] or Qol [57]. In 1 study [58], 77% of the conservatively treated patients with CLTI survived for 2 years. Vascular surgery did not improve the prognosis of the patients compared to conservative therapy; it was concluded that conservative therapy of CLTI should be considered before even attempting revascularization [58]. Patients with mild to moderate CLTI who initially received conservative treatment with optimal wound therapy and were revascularized only a few months later had similar wound healing rates, amputation rates and survival rates in the medium [59] or long term [60] as patients who immediately underwent revascularization without conservative treatment. This graded therapeutic approach may prevent unnecessary revascularization [59,60]. Often, however, patients who required primary amputation were excluded from such comparisons, which is tantamount to overestimating the success of conservative treatment of CLTI [58]. Additionally, the term “no-option” was often used very liberally.

Unfortunately, almost no study (exception: for example, 11) dealing with the prognosis of conservative therapy of CLTI differentiates between people with and without diabetes. People with diabetes mellitus have little or no pain despite ischemic foot lesions due to diabetic polyneuropathy. The adherence to pressure relief, restriction of walking distance etc. is generally low, in contrast to non-diabetic CLTI patients who voluntarily relieve their feet due to severe pain. This is why recurrent ulcers in DFS patients occur at about 30% within one year after entering remission [61]. The mortality of people with diabetes and no-option CLTI is significantly increased compared to non-diabetic persons, with comparable leg preservation rates [11].

In diabetic patients without polyneuropathy, long-term conservative treatment without escalation of the painkiller dose is not possible, which is why a major amputation should be considered sooner [14].

### 5.14. Timely Amputation

No-option DFS patients often have to offload the foot (Figure 3) in the long-term. Major amputation should be considered timely [62] in order to help patients to mobilize. Transtibial amputation, in particular, enables rapid prosthesis fitting, allowing deambulation sooner than without amputation. The longer the decision for amputation is delayed, the less likely is a benefit from a prosthetic fitting. That major amputation reduces life expectancy compared to successful revascularization [63,64] does not play a major role in this context, as mortality in patients undergoing conservative treatment is at least 22% within 1 year [65], and up to 80% within 5 years after a major amputation [3].

The influence of foot lesions or Charcot osteoarthropathy on QoL can be as negative as a major amputation [66]. QoL can significantly improve after transtibial amputation [67]. The success of the rehabilitation of major amputees was independ from whether a prosthesis could finally be worn or whether the patient remained in wheelchair [68]. Many patients on conservative therapy of CLTI have significantly improved QoL after one year, even though their state of health has not improved [63]. Getting used to the situation seems to play a major role.

A special item is revascularization of CLTI patients on dialysis. The prognosis for wound healing and survival is poor, so early amputation in patients with extensive soft tissue damage (Figure 3) and severe cardiac disease is advocated [69]. Most of these patients suffer from diabetes mellitus. In CLTI patients on dialysis with tissue loss, the absence of a revascularization option and a higher Wound, Ischemia, and foot Infection (WifI) stage were associated with lower major amputation-free limb salvage [70]. After amputation, mortality exceeds 70% at 5 years for all patients with diabetes, but 74% at 2 years for those receiving renal-replacement therapy [71].

## 6. Limitations

A major limitation of our clinical review arises from the fact that our literature research was not strictly systematic as required by the PRISMA statement guidelines for systematic reviews and meta-analyses. We chose a pragmatic literature research sufficient for a clinically oriented narrative review, since we did not expect surprisingly new information from additional literature which mostly lacks high scientific evidence. Nevertheless, the algorithm used for our research was the basis for the German evidence-based clinical practice guideline on the diagnosis and treatment of PAD, which is an internationally respected guideline of high scientific quality.

## 7. Conclusions

In DFS patients with CLTI, revascularization should be performed quickly in order to avoid major amputation (“time is leg”).

In patients with no-option CLTI, the effectiveness for gene and stem cell therapy, hyperbaric oxygen, CT-guided sympathectomy, urokinase, prostanoids, intermittent pneumatic compression, transcutaneous electrostimulation, interactive wound dressings, and the arterialization of deep leg veins remain unproven.

Conservative treatment for no-option CLTI includes the therapy of cardiovascular risk factors and concomitant diseases, appropriate wound treatment, pressure relief, avoidance of progressive soft tissue infection and timely palliative pain management. This approach may be more successful than previously assumed, especially in neuroischemic DFS patients.

For most patients who come too late for revascularization or are too sick for it, only conservative treatment or major amputation remain as therapeutic options. In certain cases, some of the above-mentioned therapeutic alternatives may be helpful. If amputation is impossible due to concomitant diseases, palliative pain therapy should be the main focus.

In no-option CLTI, timely amputation should be considered an opportunity, not an inevitable stroke of fate.

New treatment strategies for no-option CLTI patients are urgently needed. There is need for future scientific research in order to overcome the lack of efficient therapeutic options for these patients whose risk of major amputation is high.

## Figures and Tables

**Figure 1 jcm-11-02155-f001:**
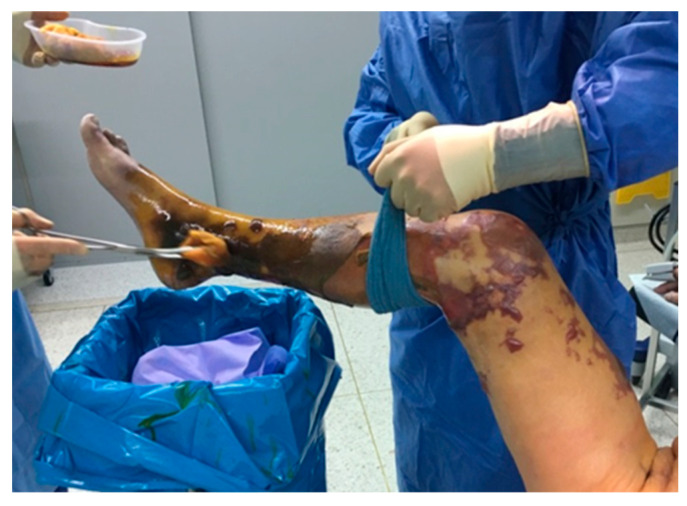
“Too late” for revascularization: extensive necrosis of the right leg.

**Figure 2 jcm-11-02155-f002:**
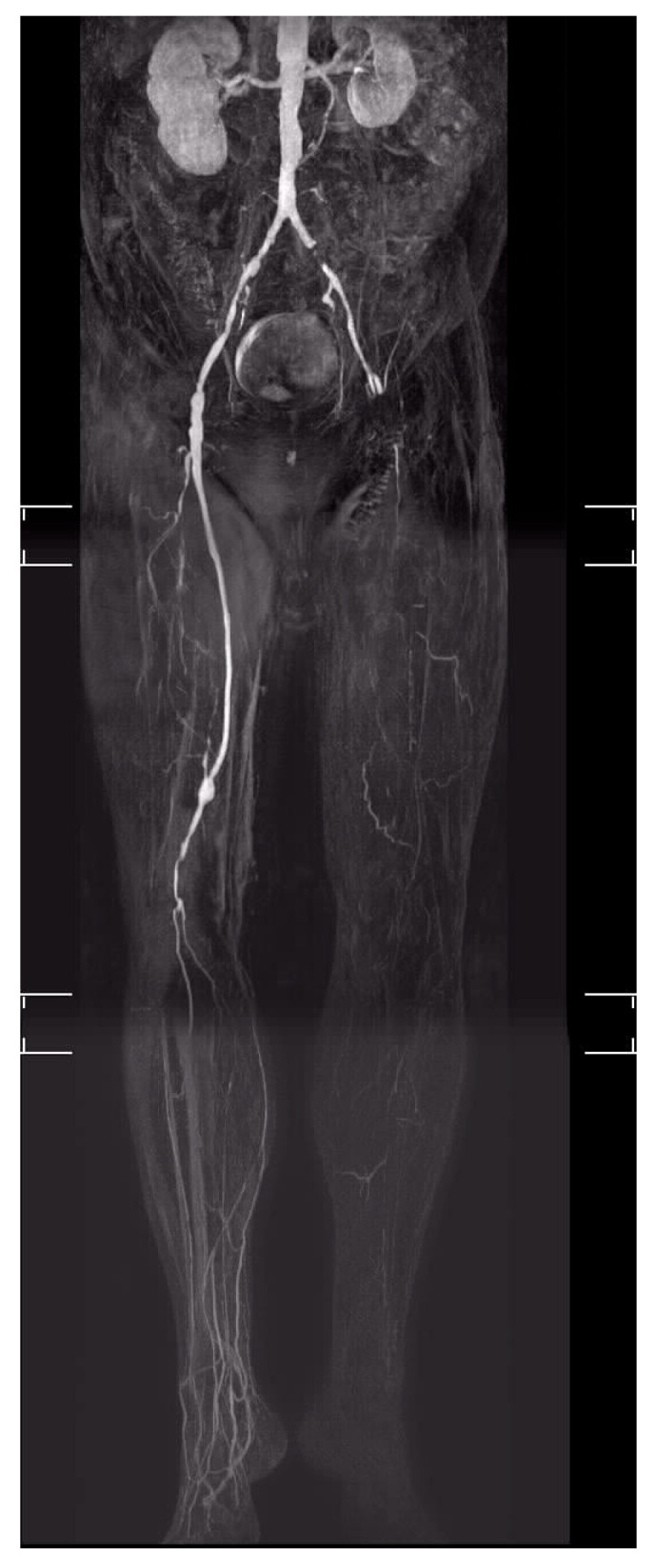
“No-option” CLTI of the left leg in a patient with DFS (Diabetic foot syndrome).

**Figure 3 jcm-11-02155-f003:**
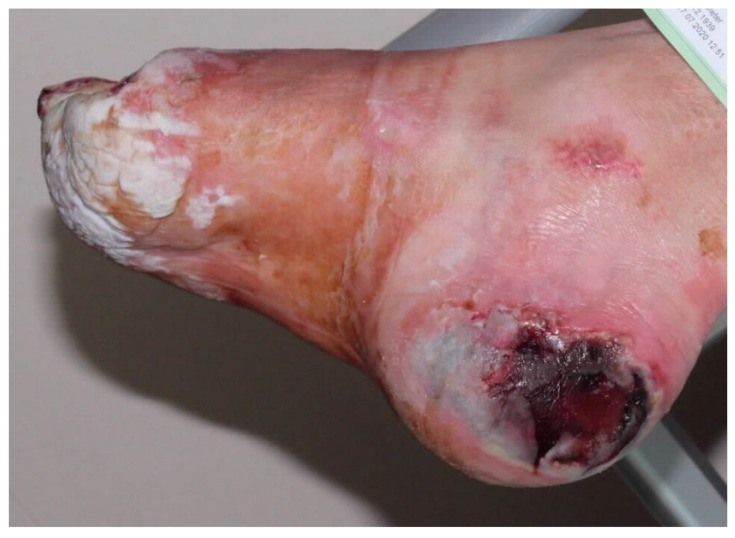
Neuro-ischemic diabetic foot in a no-option situation. The heel necrosis has increased over 4 weeks despite optimal conservative therapy.

**Table 1 jcm-11-02155-t001:** Classification of peripheral arterial disease according to Fontaine stages and Rutherford categories.

Fontaine		Rutherford		
Stadium	Klinisches Bild	Grade	Category	Clinical Picture
I	asymptomatic	0	0	asymptomatic
II a	walking distance > 200 m	I	1	mild intermittent claudication
II b	walking distance < 200 m	I	2	moderate intermittent claudication
		I	3	severe intermittent claudication
III	ischemic rest pain	II	4	ischemic rest pain
IV	ulcer, gangrene	III	5	minor tissue loss
		III	6	major tissue loss

Chronic limb-threatening ischemia (CLTI) covers stages III and IV or categories 4–6.

## Data Availability

Not applicable.

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
