# Peer review of "Therapeutic Alternatives in Diabetic Foot Patients without an Option for Revascularization: A Narrative Review"

_jcm, 2022, doi:10.3390/jcm11082155_

Round 1
Reviewer 1 Report
Authors present an interesting and complete overview about “no-treatable” patients with chronic limb ischaemia and ischaemi/neuro-ischaemic diabetic foot ulcers.
The topic is well described and analyzed such as all the barriers which reduce the chance of limb salvage and the not standard therapeutic options.
I have only minor comments:
Regarding the epidemiology of “no-option CLI patients” please cite also recent data published in JCM (Meloni M, Izzo V, Da Ros V, Morosetti D, Stefanini M, Brocco E, Giurato L, Gandini R, Uccioli L. Characteristics and Outcome for Persons with Diabetic Foot Ulcer and No-Option Critical Limb Ischemia. J Clin Med. 2020 Nov 21;9(11):3745. doi: 10.3390/jcm9113745. PMID: 33233329; PMCID: PMC7700155).
In the section “no-option” it may be interesting to understand why certain patients are very difficult to treat or revascularization is technically unfeasible or unsuccessful. One current theme is that there is a new challenging pattern in CLI diabetic patients, with more and more involvement of below-the-ankle arterial disease (approximately 40-50%) with higher risk of revascularization failure due to the higher difficult foot arteries in comparison to leg arteries. This point may be useful for readers to better understand which patients (and what patter of peripheral arterial disease) are technically more hard to be managed. Please cite Meloni M, Izzo V, Giurato L, Gandini R, Uccioli L. Below-the-ankle arterial disease severely impairs the outcomes of diabetic patients with ischemic foot ulcers. Diabetes Res Clin Pract. 2019 Jun;152:9-15. doi: 10.1016/j.diabres.2019.04.031. Epub 2019 May 9. PMID: 31078668)
Author Response
Thank You for the helpful literature! We have respected these references in the text (Lines 73 following and lines 97 following). Both publications are part of the new reference list (ref. 15 and 20).
- Meloni M, Izzo V, Da Rod V, Morosett D, Stefanini M, Brocco E, Giurato L, Gandini R, Uccioli L. Characteristics and outcome for persons with diabetic foot ulcer and no-option critical limb ischemia. J Clin Med 2020 Nov 21;9(11): 3745 DOI: 10.3390/jcm9113745. PMID:33233329; PMICD: PMCX7700155
- Meloni M, Izzo V, Giurato L, Gandini R, Uccioli L. Below-the-ankle arterial disease severely impairs the outcomes of diabetic patients with ischemic foot ulcers. Diab Res Clin Pract 2019;152:9-15 DOI: 10.1016/diabres.2019.04.031
Reviewer 2 Report
The systematic literature research demonstrated the importance of revascularization in patients with diabetic foot ulcer. For those who are impossible to receive the procedure, alternative treatments are lack of evidence of limb-preserving.
The strengths of the manuscript included the detailed literature review. The small weakness of manuscript is: some patients with limb-threatening DFU were under ESRD status with severe calcification of blood vessel. There is only one small paragraph which mentioned about the special patient group.
Only some minor point that might confuse :
1. Line 35: more than "13,000" ?
2. Line 46: "is"
3. Table 1: "clinical" picture
4. Line 70: "480,000"?
5. Line 84: d"e"bridement
6. Line 123: "Amputation-free survival was 69 %"
Is it misspelled?
Author Response
We thank the reviewer for his advice to add some more information on patients with CLTI and end stage renal disease (lines 270 following of the revised manuscript). Two additional references were cited (ref. 70 and 71 of the revised manuscript, see below). Most of these patients suffer from diabetes mellitus. In CLTI patients on dialysis with tissue loss, the absence of a revascularization option and a higher Wound, Ischemia, and foot Infection (WifI) stage were associated with lower major amputation-free limb salvage (70). After amputation, mortality exceeds 70% at 5 years for all patients with diabetes, but 74% at 2 years for those receiving renal-replacement therapy (71).
- Sigl M, Noe T, Ruemenapf G, Kraemer BK, Morbach S, Borggrefe M, Amendt K: Outcomes of severe limb ischemia with tissue loss and impact of revascularization in haemodialysis patients with wound, ischemia, and foot infection (wifi) stage 3 or 4. Vasa 49(1): 63-71, 2020. PMID: 31483747, DOI: 10.1024/0301-1526/a000819.
- Lavery LA, Hunt NA, Ndip A, Lavery DC, Van Houtum W, Boulton AJ: Impact of chronic kidney disease on survival after amputation in individuals with diabetes. Diabetes Care 33(11): 2365-2369, 2010. PMID: 20739688, DOI: 10.2337/dc10-1213
Minor points:
With respect to minor changes, we have addressed the helpful advices of the reviewer and changed the text accordingly
- Line 35: 13.000 replaced by 13,000
- Line 46: “ist“ replaced by „is“
- Table 1: “CLTInical picture“ replaced by „clinical picture“
- Line 70: 480.000 replaced by 480,000
- Line 84: “débridement“ replaced by “debridement“
- Line 123: „Amputation-free survival was 69 %: In our opinion there is no problem with this sentence, no misspelling.
Reviewer 3 Report
The manuscript entitled “Therapeutic alternatives in diabetic foot patients without an option for revascularization” aims to investigate through systematic literature review, on an alternative method of treatment in no-option CLTI patients.
Congratulations to the authors for devoting so much time and effort to conducting such a thorough literature study on CLTI.
However, my main concern regards, methodology of the systematic review itself.
Authors should follow PRISMA statement guidelines for systematic reviews and meta-analyses.
Inclusion and exclusion criteria of the articles (for instance: language, type of research paper, etc.)
In the methods section, authors should state who evaluated the studies that were considered eligible and how they were analysed.
Also, I do miss flowchart of systematic review. Furthermore, there is no result section of the systematic report, that could have summarised scientifically important data. The entire manuscript lacks a true review character; it appears to be divided into sections that need to be summed up by reporting numbers and analysing them.
Please report also summary table of all articles, if possible.
Overall statements reported in the conclusion are not justified, for instance: “For patients who come too late for revascularisation or are too sick for it, only conservative treatment or major amputation remain”; “unproven”; “It may be more successful than previously assumed”;
To draw strong conclusions, reported data must be analysed and reported in concrete numbers.
Author Response
Response to the comments of Reviewer 3
The reviewer criticises that the problem of our paper is the methodology of the systematic review itself. The authors should follow PRISMA statement guidelines for systematic reviews and meta-analyses. Nearly all his comments refer to this obvious problem.
The reviewer is perfectly right in putting into question the methods of the literature retrieval as required for a systematic review. Unfortunately, we have announced both the review (line 15) and the literature research (line 16) as “systematic“. This is not the case. Our manuscript Is meant as a „Clinical“ or „Narrative“ Review, both of which do not require the presentation of the more rigorous aspects characteristic of a systematic review such as reporting methodology, search terms, databases used, flowchart, and inclusion and exclusion criteria. Neither was our article planned as a systematic meta-analysis. We apologize for this misunderstanding. We were not aware of the strict definition of a structured systematic literature research according to PRISMA. Our review is not „systematic“, but pragmatically based on clinical items. Its aim is to courageously discuss the unsolved clinical problem of no-option CLTI which is masked by numerous efforts to prove benefitial effects of methods which are neither evidence-based nor clinically effective. Although all these methods lack significance and prove, these therapeutic options are still widely in use because nobody dares to question them, and because there is nothing to lose in a no-option situation.
Unfortunately there are very few reviews like ours on this topic, since there are many enthusiasts stuggling against any criticism of their unproven methods. There will always be someone who feels treated unkindly. Broadly speaking, there is little to no evidence for all these methods, but nobody wants to hear this.
The literature research for the present narative review is extremely braod-based, as required for an evidence-based guideline, comprising a time span of some 40 years. A systematic literature research fulfilling PRISMA standards would, in our opinion, not deliver better results. There is practically no literature with high scientific evidence on the item of our paper. In case you find it, it proves that the respective therapeutic option is not efficient and that scientific evidence is absent.
We doubt that having respected PRISMA requirements would have changed the conclusions of the present clinical review which, on an international basis, is the first of ist kind.
The basis of our paper was the literature research for the development of the evidence-based German Guideline for the Diagnosis and Treatment of Peripheral Arterial Disease (PAD) of the German Society for Angiology 2015. This literature research algorithm is well described in the Methods section of the above guidelines (VASA) and includes the identification and study of related guidelines, their evaluation according to DELBI (Deutsches Leitlinien-Bewertungsinstrument), specific literature retrieval using Medline etc. with evaluation of the scientific evidence by experts of the guideline working group, and the additional literature research on specific items such as e.g. „ozone therapy“.
Most of the objections of reviewer 3 concern this context.
However, we feel that our statements (in our words: conclusions) are justified. We are experts in the field of patients with diabetic foot problems, and we have been working scientifically in this field for decades. In our opinion, the statement “For patients who come too late for revascularisation or rare too sick for it, only conservative treatment or major amputation remain“ is perfectly right. This is a clinical fact, described in handbooks of vascular surgery, it is commonplace. Nevertheless, we have added to this statement that in cases, some of the alternative therapeutic options under discussion, e.g. deep vein arterialization, may be helpful (line 285 following).
Round 2
Reviewer 3 Report
First of all, I would like to thank the authors for their clarification regarding the previous comments.
Surely, having removed "systemic review" from the methodology, highlights the type of manuscript in a clear and concise manner.
I do not doubt the experience of the authors in the field, but readers will have to read the conclusions on the basis of the data reported in the work and not on the basis of the skills of the individual authors.
However, the narrative is interesting, it makes an excursus on the therapeutic options of the patients that angiologists and vascular surgeons face on a daily basis. and as often happens, we find ourselves with limited or even non-existent treatment options. Therefore, I find the article interesting as already commented before.
I have one last suggestion, mention the limitations of the article and the need for future scientific research.
Author Response
Response to the additional comments of reviewer 3.
We thank this reviewer for his helpful support of our manuscript. We have added a paragraph 6: Limitations (lines 286-294).
In the conclusions (lines 313-315) we have added a statement underlining the need for future scientific research in the field of no-option CLTI.
Sir,
We were able to complete the above manuscript according to the additional comments of reviewer 3. We hope that the manuscript will now meet the requirements for publication in the Journal of Clinical Medicine.
Sincerely